

# ECuADOR—Easy Curation of Angiosperm Duplicated Organellar Regions, a tool for cleaning and curating plastomes assembled from next generation sequencing pipelines

Angelo D. Armijos Carrion[1], Damien D. Hinsinger[1,2,*] and Joeri S. Strijk[1,2,3,*]

[1] Biodiversity Genomics Team, Plant Ecophysiology & Evolution Group, Guangxi Key Laboratory of Forest Ecology and Conservation, College of Forestry, Guangxi University, Nanning, Guangxi, PR China
[2] Alliance for Conservation Tree Genomics, Pha Tad Ke Botanical Garden, Luang Prabang, Laos
[3] State Key Laboratory for Conservation and Utilization of Subtropical Agro-Bioresources, Guangxi University, Nanning, Guangxi, PR China
* These authors contributed equally to this work.

Corresponding author
Joeri S. Strijk, jsstrijk@hotmail.com

## ABSTRACT

**Background:** With the rapid increase in availability of genomic resources offered by Next-Generation Sequencing (NGS) and the availability of free online genomic databases, efficient and standardized metadata curation approaches have become increasingly critical for the post-processing stages of biological data. Especially in organelle-based studies using circular chloroplast genome datasets, the assembly of the main structural regions in random order and orientation represents a major limitation in our ability to easily generate "ready-to-align" datasets for phylogenetic reconstruction, at both small and large taxonomic scales. In addition, current practices discard the most variable regions of the genomes to facilitate the alignment of the remaining coding regions. Nevertheless, no software is currently available to perform curation to such a degree, through simple detection, organization and positioning of the main plastome regions, making it a time-consuming and error-prone process. Here we introduce a fast and user friendly software *ECuADOR*, a Perl script specifically designed to automate the detection and reorganization of newly assembled plastomes obtained from any source available (NGS, sanger sequencing or assembler output).

**Methods:** *ECuADOR* uses a sliding-window approach to detect long repeated sequences in draft sequences, which then identifies the inverted repeat regions (IRs), even in case of artifactual breaks or sequencing errors and automates the rearrangement of the sequence to the widely used LSC–Irb–SSC–IRa order. This facilitates rapid post-editing steps such as creation of genome alignments, detection of variable regions, SNP detection and phylogenomic analyses.

**Results:** *ECuADOR* was successfully tested on plant families throughout the angiosperm phylogeny by curating 161 chloroplast datasets. *ECuADOR* first identified and reordered the central regions (LSC–Irb–SSC–IRa) for each dataset and then produced a new annotation for the chloroplast sequences. The process took less
than 20 min with a maximum memory requirement of 150 MB and an accuracy of over 99%.

**Conclusions:** *ECuADOR* is the sole de novo one-step recognition and re-ordination tool that provides facilitation in the post-processing analysis of the extra nuclear genomes from NGS data. The program is available at https://github.com/BiodivGenomic/ECuADOR/.

# INTRODUCTION

Chloroplast DNA (cpDNA) has been used extensively in plant phylogenetic studies as it is maternally inherited in most angiosperms, has a low mutation rate and provides variable and informative regions over broad timescales (*McPherson et al., 2013*; *Scarcelli et al., 2016*). Moreover, cpDNA provides abundant DNA polymorphisms at inter- and intraspecific levels, providing molecular phylogenies with high resolution at different taxonomic scales (*Ruhfel et al., 2014*).

The emergence of next generation sequencing (NGS) technologies, allowing for complete organellar DNA sequencing, has promoted the use of plastome (i.e., complete cpDNA sequences) data as a major tool in phylogenomic and evolutionary analyses (*Twyford & Ness, 2017*; *Zhang, Ma & Li, 2011*), as well as an extended DNA-barcode (*Kress, 2017*). As a result, massive amounts of genome-scale data have been generated at a relatively low cost, with ever faster turnaround times (*Zhang, Ma & Li, 2011*). The availability of this expanding data volume has allowed for the exploration of chloroplast organization and chloroplast features throughout the angiosperm tree at the molecular level (*Stull et al., 2013*) and the development of novel approaches for phylogenetic studies (*De Abreu et al., 2018*). To date, for angiosperms alone, there are over 4,500 sequenced plastomes available in NCBI GenBank through the INSDC database (as of June 2019) and this number has seen an exponential increase in recent years.

A consequence of the rapid development of NGS techniques is the emerging need to handle and curate increasingly large organellar genome datasets. For example, a recent study (*Li et al., 2019*), used nearly 3,000 plastomes to reconstruct angiosperm phylogeny and evolution. Because mitochondrial genomes and plastomes are usually circular, but are represented as linear text sequences in bioinformatic pipelines and online databases, any base position can represent the start of the string sequence (see Fig. 1). Plastome sequences are typically characterized by two inverted-repeated regions (hereafter called IRs) separated by a long (LSC) and a short (SSC) single copy region (*Brázda et al., 2018*; *Palmer, 1985*; *Stadermann, Holtgräwe & Weisshaar, 2016*). Due to the replication mode of organellar DNA, each part can be found in both directions (i.e., in 5′–>3′ orientation, or in the reverse complement orientation) in the cell (*Palmer, 1985*). This organization implies that orthologous regions need to be assessed for each of these parts (Fig. 1). By general consensus and some computational constraints, most of the assembly methods
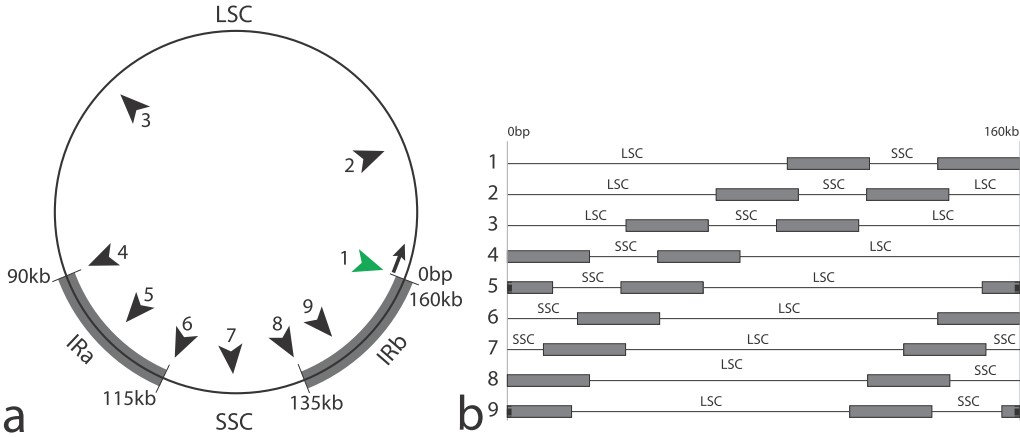

**Figure 1 Schematic representation of plastomes and effects of the artifactual linearization during the assembly process.** Schematic representation of plastomes and effects of the artifactual linearization during the assembly process. IRa and IRb, Inverted Repeats; LSC, Large Single Copy region; SSC, Small Single Copy region. (A) Circular representation showing the potential cuts (numbered black arrows) during assembly (approximate positions); green arrow: conventional start of the plastome sequence (resulting in the structure LSC–IRb–SSC–IRa); usual approximate sizes for each region are indicated. (B) Linear representations of a circular plastome, cut according to the black arrows in (A); line numbers according to (A). Note that IRs are split in three fragments in configurations 5 and 9.

start the reconstructed sequence at the first base of the large single-copy region (LSC). However, depending on both the quality and quantity of the NGS sequences (i.e., the "raw reads"), as well as the complexity of the plastome to reconstruct, a sequence may start randomly along the circular sequence (Fig. 1B), occasionally making the identification of homologous regions among sequences laborious and time-consuming, especially for large datasets (100–1,000 s of sequences).

Despite there being some consensus in the community on the starting position of plastome sequences, a quick assessment of sequences in GenBank shows extensive disparity in the organization and order of the different regions. To circumvent this, most of the plastomic studies (*Chase et al., 2016*) (including the above-mentioned 3,000 plastomes study (*Li et al., 2019*)) only consider the coding regions of the reconstructed organelles, to remove small inversions and changes in gene order. However, such rearrangements are less likely to happen at small taxonomic scales (e.g., at intra-generic level, or in species complexes). Moreover, such an approach results in the deletion of the non-coding regions, that often contain more phylogenetic signal (*Shaw et al., 2005*), especially at lower taxonomic level or in recalcitrant taxa, in which genomic approaches are usually required to fully resolve relationships (*Graham et al., 2000*; *Borsch et al., 2003*). In addition, using the entire plastome sequence allows for detection of evolutionary events, such as insertion–deletions or inversions and facilitate the identification of sequences from non-organelles origins (nuclear plastid DNA-NUPTs; nuclear mitochondrial DNA-NUMTs).

Accurate and complete molecular sequence data are essential in phylogenomic reconstruction. Therefore, accurate homology detection among sequences is a vital step.

Consequently, improvement in the functional annotation of organisms will become an extremely important step to delineate evolutionary processes.

Usually, three distinct steps can be identified in phylogenomic studies: (1) The reconstruction of organelles in the different species; (2) the assessment of orthology and alignment of the plastome sequences (i.e., the dataset assembly); and (3) the actual phylogenetic reconstruction (*Gruenstaeudl, Gerschler & Borsch, 2018*). Current software focuses on organelle reconstruction (*Castandet et al., 2016*; *Jin et al., 2018*), multiple sequence alignment (MSA) algorithms (*Bi et al., 2018*) and phylogenetic approaches (*Granados Mendoza et al., 2013*). Software currently available can aid in assembly and analyses of plastid and mitochondrial genomes. However, orthology assessment and curation of the assembled plastomes usually remains a manual task ((*McKain & Wilson, 2017*) for a combined assembly and orientation script). Indeed, there is currently no open-source software that reconstructs and assesses plastome orthology for large datasets (numbering thousands of plastomes) in a fast and accurate way from assembled draft sequences. While orthology assessment can be easily done for datasets based on individually extracted coding regions, the specific structure of the plastome (LSC, IRb, SSC, IRa) complicates the proper alignment of orthologous regions. Indeed, any differences in organization/orientation of these parts in the reconstructed draft plastomes, can result in improper assessment of orthologous positions and thus increase the risk of discarding potentially useful data at the end of alignments. This process is either done manually (representing 3–5 min per chloroplast for a trained bioinformatician, using commercially available software, for example, Geneious; *Ripma, Simpson & Hasenstab-Lehman, 2014*), or skipped by extracting the coding regions (thus discarding the most variable and useful regions of the plastome: see above). This represents an emergent bottleneck in dataset assembly and downstream analyses. Therefore, the development of flexible and user-friendly tools that remove these labor-intensive and time-consuming components from genomic workflows has become a latent priority for the plant genomics community.

We designed *ECuADOR* to facilitate both the rapid processing of organelle genomic data as well as providing output requirements for downstream analyses. *ECuADOR* is a rapid, platform-independent and user-friendly algorithm built in Perl, that automates detection and reorganization of sequence features in newly assembled plastomes. As *ECuADOR* uses only the reconstructed plastomes, it is also independent of the sequencing technology used to generate the data and can thus be used with assemblies derived from short reads (e.g., Illumina, San Diego, CA, USA), long reads (e.g., Nanopore; PacBio, Menlo Park, CA, USA) or those acquired using other sequencing technologies. Data are generated as moving singular reciprocally-compared fragments, tracking the number of nucleotide changes for a window of a user-defined length, slide along the sequence.

The algorithm is executed with default settings for the sliding window option (but with the option to adjust these manually) and adjustments can be made to input parameters according to the desired output format file (see *ECuADOR* command line below). These options can be customized via a command-line, making it user-friendly and easily accessible.

## MATERIALS AND METHODS

### Analysis pipeline

*ECuADOR* is written in Perl (tested with Perl 5.18) and uses the following modules: Bio::SeqIO, IO::String, Set::IntSpan, IO::File, Bio::AlignIO, Bio::Factory::EMBOSS, File::Temp qw/tmpnam/ and Cwd. Input for *ECuADOR* is a draft plastome sequence (GenBank or fasta), the length of the sliding window, format of input file (GenBank or fasta), and output format file (fasta or GFF3). *ECuADOR* reads and analyzes single or multiple plastomes stored in a designated folder, containing one sequence per file.

*ECuADOR* is based on a user-defined sliding window and dynamic suffix array approach. The algorithm partitions the sequence into fragment intervals, of sizes defined by the "*window size*" option. This window slides along the plastome in both 5′–>3′ and 3′–>5′ direction (as a reverse-complement sequence). Then, a positioning array index is generated from the similarity between the generated fragments. This new array index stores a sequence of the four main regions according to the exact location where each repeated extreme was found. In later stages, the extremes are used to recover the remaining IR length between both of them.

However, when the quality of the used sequencing reads is poor, mis-assemblies are likely to be introduced in the sequence during organellar reconstruction (e.g., gapped palindromes), for example, in one of the repeats. In such cases, the sliding-window approach could retrieve different sequence lengths for each repeat region. Because the main goal of the sliding-window approach is to capture the two identical-sized repeat regions (IRa and IRb), a non-equality in any repeat sequence length could prevent the recovery of its total length. This would result in an incorrect positioning for all the main regions (LSC–IRb–SSC–IRa) and prohibit the rapid assembly of correct alignments downstream. *ECuADOR* addresses this drawback by only using the extremes of each inverted repeat to estimate the repeat size, that is, once the position of each extreme fragment is known in both repeats, *ECuADOR* will take all the remaining positions between both previously positioned extremes to recover the entire repeated sequence. This offers a flexible (allowing small discrepancy between IRs-value defined as a user-customizable setting) yet conservative approach (the borders of IRs have to perfectly match each other).

*ECuADOR* allows the user to define the size of the window (in bps) used for the sliding-window step. The optimal length of the sliding window (−w parameter) depends on the quality of the reconstructed sequence. If the sequence quality is low, it is advised to use a smaller sliding window size to adjust for the higher likelihood of finding gapped palindromes throughout the inverted repeat. These would not normally be detected using a larger sliding window size, leading to a loss of information in the length of the repeats. By adjusting the sliding window size, the user can balance the sensitivity of the IR detection. A larger window size allows for reducing the influence of misassembled reads and thus false positives—at the cost of a lower resolution (− sensitivity, + specificity). A smaller window size provides increased resolution but may also increase the number of false positives if the data is noisy (+ sensitivity, − specificity). The quality of the input
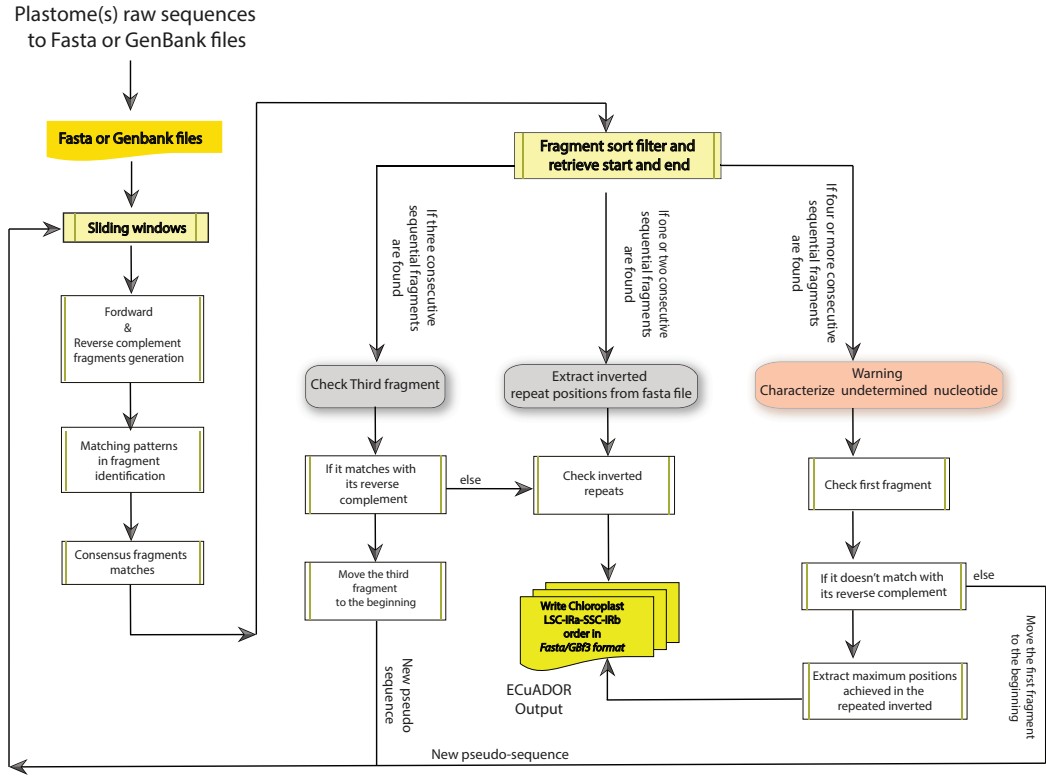

**Figure 2 Flow chart of *ECuADOR*.**

plastome sequence (e.g., as a consequence of a low quality base call during sequencing or mis-assemblies) is an important factor to take into account before setting a value for −w (i.e., the window size parameter).

Finally, *ECuADOR* is executed using the following command-line perl command: *ECuADOR*.pl (−h) −i, <folder containing the plastomes>, −w, <sliding window length> (1,000 bp option by default), −f, <input file format (option not by default, GenBank or fasta)>, −out <output prefix>, −ext, <output format> (either fasta or GFF3), −save_regions, <save chloroplast regions (LSC, SSC, IRs separately, or in combination)>, −orient <TRUE> (reorientation of each plastome regions, providing the user with the cpDNA regions ready to use for MSA), finally the option −noIRs <3 or 4> (to generate files including either only the concatenated regions LSC–IRb–SSC or the entire one LSC–IRb–SSC–IRa)

The program is divided into seven main steps (Fig. 2):

1. Argument check and analysis of the input file, setting format files and sliding window size (if no slide window size is provided, the default option will be set to 1,000 bp.

2. *ECuADOR* generates a reverse complement sequence of the input fasta sequence (i.e., reconstructed draft plastome).

3. A suffix array index (permutation of index numbers giving the starting positions of suffixes of a string in alphabetical order) is built, comparing the suffix of fragment sequences with each and every suffix of the reverse complement fragments (both generated through slide windows). This new array index will be used to detect the exact

location of the corresponding extremes to each IR. In the usual case (gap-free or identical palindromes), a total of two arrangements of only one sequential element will be located in the suffix arrays, thus covering the exact position of each repeat. In unusual cases (gapped palindromes) two arrangements of multi non-sequential elements will be located in the suffix arrays. In such cases these extra fragments will prevent the recovery of the repeated regions.

4. If three or more un-sequential elements are detected in any of the two main arrangements as a result of gapped palindromes, *ECuADOR* takes the extra smallest dissimilar sequential elements and it will create a new fragment from them. This new fragment will contain the start position of the multi non-sequential elements located in the first inverted repeat with the end position of the multi non-sequential elements located in the second inverted repeat. Then, this new fragment will be concatenated with the ending homologous fragment.

5. Once the inverted repeat regions are located, the algorithm maps the remaining positions (LSC and SSC respectively) through a sequence sweep from start to end.

6. When all the four main regions are found through mapping-by similarity-analysis (LSC, IRa, SSC and IRb respectively), *ECuADOR* reorganizes the structure of the entire sequence in the same order for all the analyzed plastomes.

7. *ECuADOR* generates five output files. The first one is a summary of the main annotation features of the regions, including the length of the IRs. The second one is the reordered sequences (i.e., LSC–IRb–SSC–IRa) in either fasta or GFF3 format, depending on user options (for GenBank files, it will provide a complete reorientation and extraction for all its established annotations in a new GFF3 output file). If a sequence cannot be reliably curated, *ECuADOR* generates a third file containing a problem description for that particular sequence. If the option "all-orient TRUE" was selected, an additional step is performed, that will homogeneize the direction of the sequences for each plastome fragment before generating the final results files (for this purpose, we integrated in the *ECuADOR* code a modified version of the script *seqOrient.pl*, available at). Finally, we included an additional option number of inverted repeats ("-noIRs") which allows the user to get either one or two IRs. The default value is 1 (thus outputting the LSC–IRb–SSC), while "-noIRs 2" will output the complete plastome sequence (LSC–IRb–SSC–IRa, useful e.g., before annotation for GenBank submission). The output (fasta file) can be immediately used in software for downstream phylogenetic and genomic analysis (e.g., Geneious, CLC workbench or MAFFT, PHYML and BEAST).

## Taxon selection and dataset construction

To test and assess performance of *ECuADOR*, two main datasets and a reference case were generated. The first (dataset1: *data control*) was used to reevaluate sequences corresponding to 161 published plastomes from a selection of 51 major angiosperm groups (*Ruhfel et al., 2014*). Fasta sequences of the plastomes were downloaded from GenBank

and analyzed using *ECuADOR* with default parameters and a sliding window adjusted to 1,500 bp. We reviewed the accuracy of IR identification by evaluating the similarity and position of each retrieved region compared to their original annotation, using Geneious R9 v.9.0.5 (*Ripma, Simpson & Hasenstab-Lehman, 2014*), with the "Find repeats" function (1,500 bp as minimum repeat length; no allowed mismatch between repeats). To further evaluate the quality of the obtained locations (LSC, SSC, IRs), we built a phylogenetic tree based on the 161 reanalyzed plastomes and compared this with the previously published topology (*Ruhfel et al., 2014*). The sequences were aligned using MAFFT v.7 (*Katoh & Standley, 2013*) under the FFT-NS-1 option and PhyML v3.3.20 (*Guindon et al., 2010*) was used to build a maximum-likelihood (ML) tree with the GTR DNA substitution model and the fast likelihood based method (aLRT SH-like).

Secondly, to assess the robustness of our algorithm, we simulated low quality/noisy plastomes by introducing random substitutions with different percentages of variation in the plastome of *Arabidopsis thaliana* (dataset2: *data testing*). Eleven levels of variation were chosen, ranging from 0.01% to 5.31% and 1,000 simulations were generated for each variation level, respectively, using an in-house script (available at https://github.com/BiodivGenomic/ECuADOR/). These "low quality" plastomes were then evaluated through *ECuADOR* with default parameters.

Finally, in order to evaluate the applicability of our algorithm to any kind of chloroplast data (not only for families previously analyzed) and taking advantage of its fast detection and extraction speed, a dataset corresponding to 4,541 angiosperm chloroplasts was downloaded from GenBank (database INSDC accessed on 2019/06/20-dataset3: *mass data evaluation*) and analyzed using default options in the same format. This further served as an additional survey to detect cases with missing or poorly uploaded regions currently available in GenBank. The goal of this analysis was to evaluate the percentage of negative cases (i.e., without any IR identified), using the largest number of chloroplasts available for angiosperms to date and to determine the underlying causes of these events.

## RESULTS

### Prediction of performance analysis (Dataset1: data control)

To evaluate *ECuADOR*, we compiled 161 plastome sequences from a total of 51 major angiosperm groups, as previously used in *Ruhfel et al. (2014)*. *ECuADOR* ran fluently for each plastome, with regions identified in almost all sequences (Table S1). It provided basic information regarding the location and inverted repeat lengths, reordering of the main plastome regions (LSC–IRb–SSC–IRa) as well as the repositioning for all the protein coding genes (CDS) in gff3 format file for all plastomes analyzed. Furthermore, it substantially eases the post-processing analyses of plastomes reconstructed from NGS data. Results obtained with *ECuADOR* and Geneious were very similar, validating the performance and accuracy of our approach (Table S1). Retrieved annotations were identical for 150 sequences out of 161. This number increased to 160 after manually setting the first position of the LSC as the start of the plastome (contrary to *ECuADOR*, in which this is done automatically). IR annotations were not retrieved accurately for one draft plastome sequence (GU592211), due to very poor sequence quality.

ECuADOR took 15 min to analyze this dataset using a MacBook Pro, 2.2 GHz Core 2 Duo, 16 Gb RAM. When compared to manual individual treatment (e.g., average of 3–5 min handling per plastome), this would have taken between 8 and 13 h.

The topology of the obtained phylogenetic tree (Fig. 3) was similar for 49 families compared to previously identified relationships (*Ruhfel et al., 2014*). An inconsistency was found in the placement of Ranunculaceae (*Ranunculus macranthus*), which grouped together with Piperaceae, Dioscoreaceae and Chloranthaceae.

## Performance of the introduced variation simulation (Dataset2: data testing)

The introduction of mismatches between both repeats is based on the loss of information, thus reducing the identity of the IRs and altering the final reorganization of the plastome. This analysis allowed us to understand how the introduced error for the different simulation sets affects the recovery of the original positions of the inverted repeats and therefore the ability of the algorithm to retrieve the reordinated sequence completely. Thus, for each mismatch level, we scored and investigated cases where *ECuADOR* failed to retrieve the original IRs locations (due to excessive variation in the base pair numbers within the inverted repeats), using as a model the *A. thaliana* chloroplast genome (Fig. 4).

*ECuADOR* was able to recover and reorder the main regions of the plastomes (LSC–IRb–SSC–IRa) for each altered dataset. As expected, the mismatching percentage affecting the true annotation increased as the alteration for each data set increased. *ECuADOR* showed an accuracy above 90% with 22 or fewer alterations (Table S2). Such a high level of mismatch between the two IRs, likely represents major misassembled positions and we recommend such low-quality draft plastomes should first be carefully checked to assess the origin of such mismatches. Moreover, the user can easily modify the stringency of the detection process by specifying the sliding window fragment size to improve search precision. This should however only be done for noisy datasets which are known to contain high levels of mismatching or highly similar fragments throughout the plastome.

## Reference data set (Dataset3: mass data evaluation)

To evaluate the potential and flexibility of *ECuADOR*, we analyzed a total of 4,541 angiosperm plastomes using default parameters with fasta format as input. The main regions were successfully detected in 4,446 sequences (97.90%, File S1) whereas identification and organization of the plastome structure failed in 95 sequences (2.09%, File S2).

To further assess the causes of these failed instances, the 95 sequences were evaluated in Geneious (*Ripma, Simpson & Hasenstab-Lehman, 2014*) and we found manual curation was needed to identify the main regions in these plastomes. Indeed, these plastomes had missing regions or poorly formatted annotations in GenBank, or were of extremely poor quality. Finally, several plastomes were characterized by an absence of inverted repeats (e.g., gymnosperm plastomes and several species in parasitic plants (*Wicke et al., 2013, 2016*)).

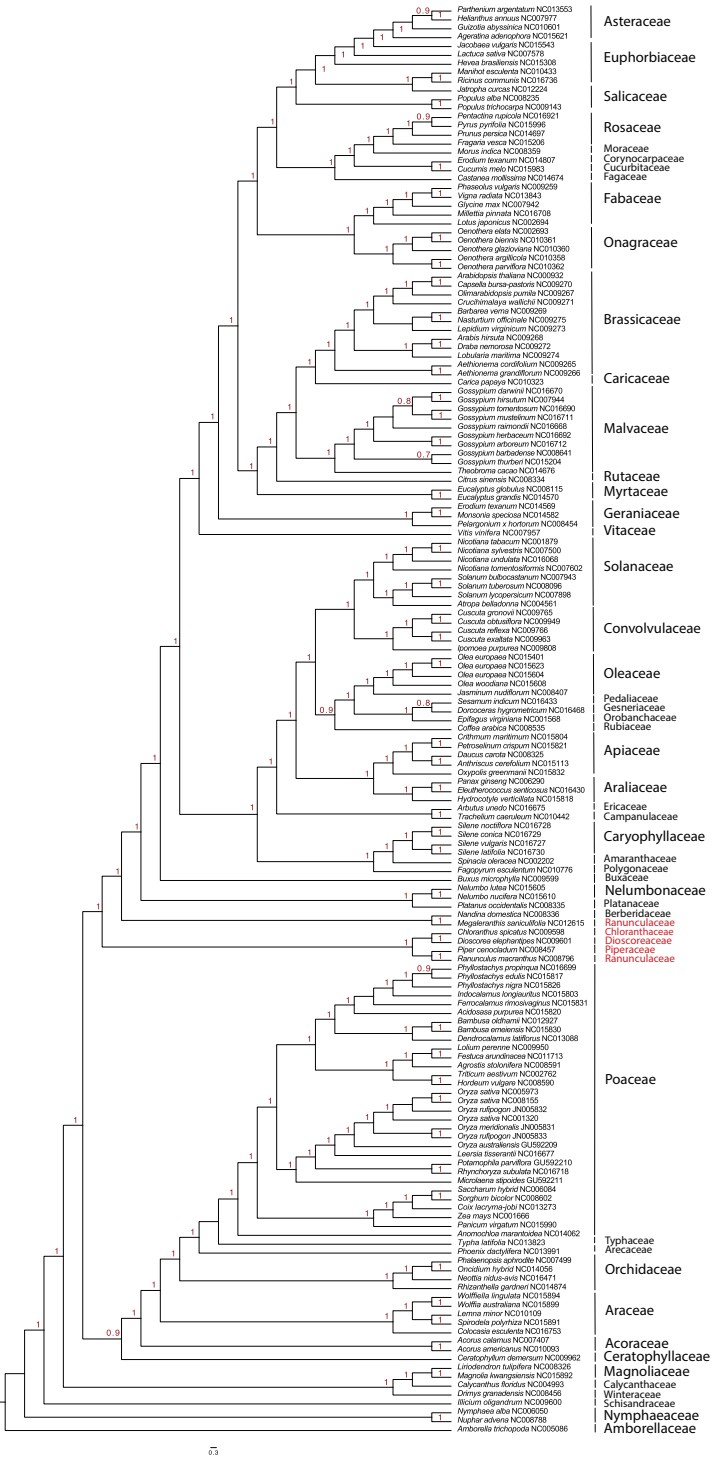

**Figure 3 Phylogenetic tree constructed with 161 cpDNAs, using fast likelihood-based method (aLRT SH-like) as implemented in PhyML (*Guindon et al., 2010*).** Numbers on nodes indicate probability values. Families highlighted in red show an inconsistency found in the placement of Ranunculaceae (*Ranunculus macranthus*), which groups together with Piperaceae, Dioscoreaceae and Chloranthaceae.

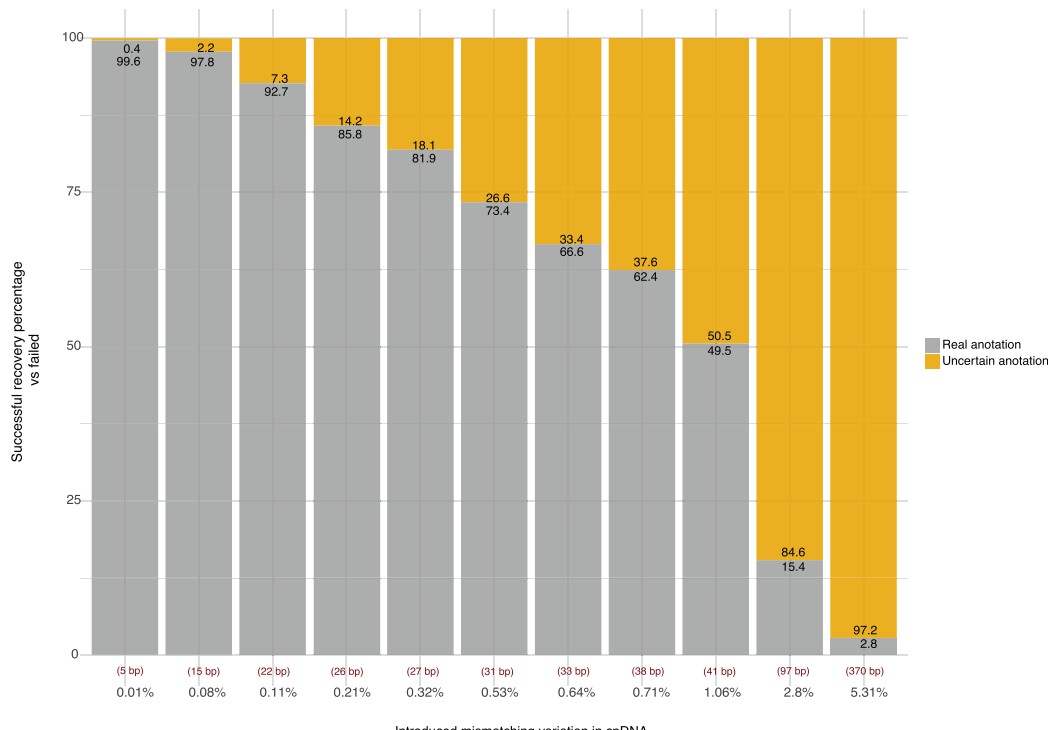

**Figure 4 Accuracy of ECuADOR in retrieving the correct IR locations in plastomes of decreasing quality.** Vertical axis percentage of simulations where correct (grey) or incorrect (yellow) IR locations were retrieved. Horizontal axis: percentage of mismatching positions introduced in the IR sequences of the *Arabidopsis thaliana* reference plastome sequence (NC_000932) for 1,000 simulations. Values in the lower part shows the total assigned variation in base pairs for each set respectively. Red values below the bars show the error average in base pairs for the positioning of the uncertain annotation.

ECuADOR requires a correct match between both repeats to be able to recover the newly generated regions (LSC–IRb–SSC–IRa) and score the analysis as successful. The script will not identify IRs in sequences of extremely poor quality. Nevertheless, it has several detection error and correction mechanisms—such as a wide detection of more than one inverted repeat region and recognition of ambiguous characters in the plastome.

## DISCUSSION

*ECuADOR* is a novel algorithm designed for the identification, reorganization and reordering of homologous regions (LSC, SSC and IRs) on large-scale plant datasets using the specific location of the inverted repeats in any point of the circular genome as a starting point. Other software aimed at helping with circular genomes exist, but none handles the post-assembly curation in a standardized manner. For example, *Circlator* (*Hunt et al., 2015*) was designed to use with long sequencing reads (e.g., PacBio data, Menlo Park, CA, USA) in bacterial chromosomes and plasmids and the plastid and mitochondrial genomes of eukaryotes to reconstruct circular genomes. However, *Circlator* is an assembler and thus works with the filtered, long sequencing reads (FASTQ files) to generate a circular genome. In that way, it is not different from the many assemblers or

scripts specifically designed to generate organelles genomes (e.g., ORG.asm, NovoPlasty, GetOrganelle) that can be used before *ECuADOR*. As most phylogenomic studies combine both new sequences and plastomes mined from GenBank, original raw data can be missing for a significant part of the sampling. Although long sequencing reads are quickly emerging as a powerful tool in genomics, the vast majority of generated data available are short reads (<300 bp) from Illumina platforms, prohibiting the use of a long-read-specific solution. As *Circlator* could be involved in the assembly process, *GenomeRing* (*Herbig et al., 2012*) is a visualization tool that allows an overview of several plastomes in the same coordinate system. In that sense, it could be considered as a good complement to *ECuADOR*, to visualize the results after alignment. But given that an alignment is a requirement for *GenomeRing* and the program does not generate any files for immediate downstream analyses, we cannot consider *GenomeRing* as an alternative for *ECuADOR*. The closest algorithm to *ECuADOR* would be *MARS* (*Maurer et al., 2005*), as it can homogeneize the starting point of a set of sequences from a circular genome, a function *ECuADOR* also performs as a side-effect of the reorganization of the plastome. However, *ECuADOR* is able to not only move the starting point, but also to keep order and orientation of the main regions of the plastome. In addition, *MARS* outputs a FASTA file, while *ECuADOR* can keep track of all annotations previously included in the input files. Fast-Plast (*McKain & Wilson, 2017*), despite being a "all-in-one" pipeline including assembly and ordering of the plastome, is not able to handle GenBank sequences, limiting its use to newly generated draft plastomes. In addition, its structure implies to work with each plastome separately, while *ECuADOR* is able to homogeneize the output order and direction for a complete set of sequences, making downstream analyses easier.

A possible explanation for the incongruence observed in dataset1 could be that *Ruhfel et al. (2014)* used only protein coding data of 78 genes from 360 taxa to build their phylogenetic tree. Gene conflicts in plastome-based phylogenies have recently been highlighted as a major cause of incongruence among studies (*Gonçalves et al., 2019*; *Walker et al., 2019*), and differences in methodology between our study and the study by *Ruhfel et al. (2014)* could explain the observed incongruence. The analyses performed here were based on complete plastome data and include non-coding, fast evolving regions. Despite that these regions have been proven useful at smaller taxonomic scales, they are expected to saturate at very large scales, resulting in nonspecific phylogenetic signals in deeper parts of the tree and causing incongruence with signals inherent in coding-regions. It is beyond the scope of our study to provide a detailed analysis of the problems inherent in the reconstruction of the phylogeny of angiosperms but worthy to note that the level of saturation effects in the plastome dataset seems low, with only one incongruence compared to the analysis using coding-regions only.

The introduction of *ECuADOR* has provided a major step forward in our ability to quickly identify and extract the main plastome regions in a coordinated, standardized arrangement. This new reference system not only will allow to define a global reorganization for the main plastome regions but can also be employed to generate a chain

of re-repositioning for all the remaining GenBank annotations in the sequence. This last condition applies to all plastomes in GenBank in which the string starts randomly along the circular sequence. This allows to recast the previous reference annotation into a new coordinate system for all the available annotations and print this out in a GFF3 output file. In addition, the generation of reorganized and reoriented datasets, thus already formatted for downstream analyses (e.g., alignment) will greatly improve the utility of plastomes, either newly sequenced or mined from GenBank. Finally, the new coordinate system can be used to implement exploratory methods for a more accurate and faster analysis of phylogenetic comparative data, either using complete regions or concrete molecular markers in case of using GenBank files. This, in turn, will allow us to advance the development of more accurate hypotheses in the reconstruction of the evolutionary history of extant plant groups.

## CONCLUSIONS

Curating draft plastomes and formatting them for downstream phylogenetic analyses is laborious, time-consuming and error-prone. We developed *ECuADOR* for the robust extraction and mass reorganization of plastome regions. The proposed algorithm is based on sliding windows and dynamic suffix array approaches to track inverted repeat locations, followed by extraction and repositioning of the main chloroplast regions. In addition, the user can generate datasets in which all sequences are similarly oriented, allowing a direct inference of the homology through sequence alignment. This facilitates post-processing analyses of extra-nuclear genomes from NGS data, optimizing handling times and reducing error. We demonstrated its accuracy, especially in handling poorly reconstructed plastomes, when repeats are interrupted by misassembled positions (resulting in fragments poorly positioned throughout the sequence), preventing recovery of IRs. This method significantly reduces handling time and complexity in the analysis of large plastome datasets and allows for error free processing of high quantities of data. Our study not only underscores the importance of developing new tools for detecting and characterizing inverted repeated sequences, but also provides a new approach to systematically identify complete regions within plastomes. *ECuADOR* will be maintained and regularly improved to add new features, according to the emergence of new needs with the development of innovative approaches in NGS. For example, future scheduled improvements include the extraction of the homologous genes and non-coding regions (i.e., intergenic spacers, introns and ribosomal RNA) to generate locus-specific alignments, as currently widely used to avoid the laborious steps involved in manual curation. We believe *ECuADOR* has the potential to be useful and widely applicable to the plant science community that handles large genomic datasets on a daily basis, whether this is for genomics/phylogenomics, evolution, ecology or bioinformatics.

## ACKNOWLEDGEMENTS

We kindly thank F. Areces-Berazain for providing useful comments to an earlier version of this manuscript.

### Funding

This work was supported by grants from Guangxi University (Nanning, PR China), the State Key Laboratory for Conservation and Utilization of Subtropical Agro-bioresources (GXU, Nanning, PR China) and the Bagui Scholarship team funding under Grant No. C33600992001 to Joeri S. Strijk, as well as a Guangxi University grant (XDZ120929) and China Postdoctoral Science Foundation Grants (Nos. 2015M582481 and 2016T90822) to Damien D. Hinsinger. The funders had no role in study design, data collection and analysis, decision to publish, or preparation of the manuscript.

### Grant Disclosures

The following grant information was disclosed by the authors:
Guangxi University, Nanning, PR China.
State Key Laboratory for Conservation and Utilization of Subtropical Agro-bioresources, GXU, Nanning, PR China.
Bagui Scholarship: C33600992001.
Guangxi University: XDZ120929.
China Postdoctoral Science Foundation: 2015M582481 and 2016T90822.

### Competing Interests

The authors declare that they have no competing interests.

### Author Contributions

- Angelo D. Armijos Carrion performed the experiments, analyzed the data, prepared figures and/or tables, authored or reviewed drafts of the paper, designed the algorithm, wrote the program, and approved the final draft.
- Damien D. Hinsinger conceived and designed the experiments, authored or reviewed drafts of the paper, designed the algorithm, and approved the final draft.
- Joeri S. Strijk conceived and designed the experiments, authored or reviewed drafts of the paper, and approved the final draft.

### Data Availability

The program is available at GitHub: https://github.com/BiodivGenomic/ECuADOR/

### Supplemental Information

Supplemental information for this article can be found online at http://dx.doi.org/10.7717/peerj.8699#supplemental-information.

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
