# Peer review of "ECuADOR—Easy Curation of Angiosperm Duplicated Organellar Regions, a tool for cleaning and curating plastomes assembled from next generation sequencing pipelines"

_PeerJ, doi:10.7717/peerj.8699_

## Round 0.1 · original submission · Major Revisions

Together the three reviewers have picked up a substantial number of issues that need to be adressed. Please work through this constructive feedback and incorporate the suggestions.

Reviewer 1 ·

Basic reporting

I found that the paper is well written and reports a bioinformatics tool that could be useful in large phylogenomics studies, especially using available data on GenBank. A do have a few corrections to suggest both to the text and the code.
While seems counter-intuitive the usual order of plastome is reported as LSC-IRb-SSC-IRa. Please check the paper for OGDRAW (https://doi.org/10.1093/nar/gkz238) and references 21, 22 within for the background of this. I believe following this would only take switching IRa and IRb labels in your code.
I think you should mention Fast-Plast in your introduction, as after short read assembly it outputs the plastome in the canonical LSC-IRb-SSC-IRa order, but cannot handle already assembled GenBank data. RepeatFinder in Geneious could also be mentioned as this will identify the inverted repeats but any necessary rearrangements would still need to be done by hand.
It might be useful, if possible, to give the option to output the assembly as LSC-IR-SSC, i.e. without the second IR copy. This could allow easy data formatting for phylogenetic analysis to avoid overestimating the signal from the IR.
Experimental design
Sequence data submitted to GenBank are often included in two source databases: INSDC (GenBank) and RefSeq. RefSeq accession numbers normally start with NC_. This means that your Dataset3 contains duplicated sequences e.g. NC_030299 and KU351660 are the same Ziziphus jujuba assemblies. I recommend that you rerun the Dataset3 analysis with only the INSDC (GenBank) records to get a more accurate success rate.

Experimental design

Sequence data submitted to GenBank are often included in two source databases: INSDC (GenBank) and RefSeq. RefSeq accession numbers normally start with NC_. This means that your Dataset3 contains duplicated sequences e.g. NC_030299 and KU351660 are the same Ziziphus jujuba assemblies. I recommend that you rerun the Dataset3 analysis with only the INSDC (GenBank) records to get a more accurate success rate.

Validity of the findings

While I believe your work will be useful I think it contains some errors in reporting the length and starting positions in the inverted repeats. As far as I can see the length is calculated by subtracting the start position from the stop. This results in 1 bp shorter IRs than it should be as subtraction is inclusive only on one end. I suggest either subtracting the stop position of LSC or SSC from the stop positions of the IRs or just simply add one to the existing calculations.
If assemblies are correct the end of the sequence, i.e the 3’ end of IRb in your paper, should match the 5’ end of IRa. I randomly checked this in ten sequences, one of which I believe is a poor assembly. The start position of IRa in the other nine is 1 bp off from what is suggested by the end of the whole assembly. I think it would be useful to run another analysis to compare the recovered IRs between your script and RepeatFinder or any other programmes that you can think of.
Moreover, length of the IRb calculated as the difference of stop and start position does not match the reported length in nine out of the ten sequences.

Additional comments

Line 71: It is ambiguous what to date is. I would recommend including the date sequences were downloaded, just like in Dataset3
Line 72: I think7,000 is incorrect due to RefSeq duplicated sequences
Line 80: I think inverted repeats can only be palindromic if there are no intervening bases between the two parts, which is not the case for the plastome. I think it would be simpler to just use inverted repeats and IRs for short.
Line 81-83: I think you are describing the isomeric copies of the plastid genome, which should have the original citation (10.1146/annurev.ge.19.120185.001545) and probably some of the newer ones where isomeric copies were mistakenly thought to be inversions (doi:10.3732/ajb.1500299).
Line 84: I am not entirely sure what each of these refer to
Line 93-94: This is really only one of the reasons, though probably the most common one. With small inversion and changes in gene order the non-coding intergenic spacers require laborious checks for homology.
Line 100: I think this should be non-organelle origins and the NUPTs and NUMTs abbreviations should be resolved in the first mention.
Line 167-168: This suggest that by default the shorter repeat will be taken as correct, while this is not necessarily correct.
Line 245: I think it would be useful to make the in-house script available on github
Line 268: There is an extra bracket here
Line 278 & 287: I found these difficult to follow. It might be simpler that accuracy was above 90% with 22 or fewer alterations.
Line 324: This should be plastome data
Line 387: It might be better to use large genomic datasets

Reviewer 2 ·

Basic reporting

Reporting seems well done, please see general comments.

Experimental design

Experimental design seems well done, please see general comments.

Validity of the findings

Validity seems well done, please see general comments.

Additional comments

This paper describes a new program for finding orthology among regions in the plastome. This includes not only coding sequences, but can also identify non-coding sequences. Essentially it feels like the now defunct program DOGMA (Wyman et al. 2004), with phylogenetic data assembly step. From what I can tell this is a valuable niche to be filled, and although some programs do similar procedures, it’s always good to have multiple programs that do something and this is the only one I am aware of specifically capable of combining new data and genbank data.

Wyman, Stacia K., Robert K. Jansen, and Jeffrey L. Boore. "Automatic annotation of organellar genomes with DOGMA." Bioinformatics 20, no. 17 (2004): 3252-3255.


Throughout the paper it would be good to have another run through for grammar and spelling, I did not notice any major issues but I believe it could benefit from another examination.

One thought and this is strictly up to the authors, I don’t by any means is necessary for publication, is that it would be good for the program to also be able to combine with large transcriptome datasets and add in the coding sequences from those. Transcriptomes seem to be an abundant source of this chloroplast material also and although it does not provide the non-coding DNA being able to complement the coding from chloroplasts and the non-coding may be valuable.

Just a quick note it seems as though for the phylogenetic analysis I would imagine it would be bad to include both IRa and IRb as this represents redundant data. This could be something to ensure is mentioned in the paper or on the authors Github to avoid this mistake being made. One of the issues with mass processing is the ability for simple errors such as that to emerge.


At line 340 the discordance between this and the work of Ruhfel et al. makes sense based on studies that are finding conflict among genes and dataset assembly such as Goncalves et al. 2019 and Walker et al. 2019.

Gonçalves, Deise JP, Beryl B. Simpson, Edgardo M. Ortiz, Gustavo H. Shimizu, and Robert K. Jansen. "Incongruence between gene trees and species trees and phylogenetic signal variation in plastid genes." Molecular phylogenetics and evolution (2019).

Walker, Joseph F., Nathanael Walker-Hale, Oscar M. Vargas, Drew A. Larson, and Gregory W. Stull. "Characterizing gene tree conflict in plastome-inferred phylogenies." PeerJ 7 (2019): e7747.


I believe the authors have done a good job finding an area where there is clearly a need for a program and putting together a good program to fulfill that need.

Reviewer 3 ·

Basic reporting

no comment

Experimental design

no comment

Validity of the findings

no comment

Additional comments

In this study, the authors present software for the detection and organisation of chloroplast regions across multiple assembled sequences, streamlining analyses that are normally time consuming and potentially prone to error. As far as I am aware, I believe this software is novel in that in removes the necessity for manual correction and will likely be of value to future studies in chloroplast NGS studies.

I think that the manuscript is well written and structured clearly, and software testing was comprehensive and informative for potential users. With some very minor changes I feel this should be accepted for publication and I look forward to trialing it on my own data.

I only have minor comments on the manuscripts as outlined below.

Lines 71-73 – when and how was the number of plastomes on GenBank accessed?

Lines 93-95 – add references to “most of the plastomic studies” and “above mentioned”

There were a couple of acronyms that I didn’t understand. It might be worth spelling these out in full. Line 100 – NUPTs and NUMTs. Line 120 – BGT.

Line 114 – I thought the line beginning “Despite that it can usually be done” sounded odd but this might just be me. Perhaps change to “While reconstruction can be easily done for…”

Line 179 – “The amount of low quality in the original data…”. Perhaps specify “low quality sequences” or “low quality base calls” for example.

I thought it made sense to refer to the sliding window parameter “-w” on line 170, at the start of the paragraph, rather than on line 180.

Line 245 – Am I correct in thinking random substitutions introduced as way of creating noisy/low quality datasets? If so, I thought it would be good to state this clearly for the reader.

In figure 3, refer to how the tree was generated in the figure heading, and detail all the usual information such the values on the branches and branch length scale. Also, specify why some families are highlighted in red. I realise this is discussed in text but it would be good to clarify in the figure heading too. The size of the text is quite small, so I expect the quality of the figure will need to be good for publication.

---

## Round 0.2 · Minor Revisions

Providing you deal with the remaining revisions suggested I would not expect to send this out for any further review before acceptance.

Reviewer 1 ·

Basic reporting

The article has been improved greatly. I have very few comments below mostly suggestions to improve the text. I would be happy to see the paper published after these are incorporated.

Experimental design

no comment

Validity of the findings

no comment

Additional comments

Line 47, 53, 171: Irb should be changed to IRb
Line 83-84: I think ref 10 should be cited for this sentence.
Line 94: It should be quick assessment of sequences
Line 99: delete of
Line 106: I think “Accurate and complete molecular sequence data are essential in phylogenomic reconstruction” could be easier to read
Line 110: There is an extra in
Line 117 & 365: Fast-Plast does not do annotation, only assembly and orientation
Line 122: potentially might be better here
Line 176: It should be a not an
Line 192 & 194: There are extra spaces at some of the < or >
Line 249: I am not sure what you mean known plastomes, but I don’t think it is needed here
Line 264: I think it would be useful to include the GitHub link for the script here
Line 272: I don’t understand possible cases without any detection. Could you rephrase this?
Line 349: This should be plastome and you can delete of
Line 415: If you are planning to further develop your software to generate gene alignments I think it would be useful to have the option of getting alignments of homologous non-coding regions as well. I don’t think this is needed for the paper but I thought it would be worth mentioning it.

Reviewer 2 ·

Basic reporting

The authors present the revision of Ecuador a program to facilitate the preparation of plastome data for downstream analyses. They have done a good job addressing all comments from the first round of review and I believe this will be a useful tool in the ever-growing field of plastome phylogenetics.

Experimental design

I don't see any issues in the Experimental design

Validity of the findings

All findings appear valid.

Additional comments

At several places in the paper you have the “r” in SSC-Irb-LSC-IRa in lowercase, I believe it should be uppercase. And at line 226 you have Ira instead of Ira.

Reviewer 3 ·

Basic reporting

NA

Experimental design

NA

Validity of the findings

NA

Additional comments

I believe the authors have largely addressed the reviewer comments. I only have minor comments.

Lines 73-75: This is a very minor point but how was the number of plastomes on genbank accessed? I am assuming it was through the ncbi ftp database but it would be good to confirm.

Lines 97-99: The following sentence didn't make sense to me and might be worth rewording.
"However, despite such rearrangements are less likely to happen at small taxonomic scale (e.g. at intrageneric level, or in species-complexes), most of studies at various scales use only coding regions."

Line 110: delete repeated word "in"

Line 126: "for a trained worker in the Biodiversity Genomics Team (BGT) lab". Perhaps I have misunderstood but could this equally read "for a trained bioinformatician"?

Line 349: Change "platome" to "plastome"

Figure 3: cite that you used PhyMl software in the figure heading for clarity. The quality of the figure may not be of sufficient for publication given the number of tips. In particular the support values will not be legible, but there are other ways of representing support values on a tree, for example colouring the nodes to represent support.

---

## Round 0.3 · accepted · Accept

Thank you for incorporating all the corrections and suggestions of the reviewers.